# Thermo-Responsive PLGA-PEG-PLGA Hydrogels as Novel Injectable Platforms for Neuroprotective Combined Therapies in the Treatment of Retinal Degenerative Diseases

**DOI:** 10.3390/pharmaceutics13020234

**Published:** 2021-02-07

**Authors:** José Javier López-Cano, Vanessa Andrés-Guerrero, Hongyun Tai, Irene Bravo-Osuna, Irene Teresa Molina-Martínez, Wenxin Wang, Rocío Herrero-Vanrell

**Affiliations:** 1Research Group (UCM 920415), Innovation, Therapy and Pharmaceutical Development in Ophthalmology (InnOftal), Complutense University, 28040 Madrid, Spain; josejavl@ucm.es (J.J.L.-C.); vandres@ucm.es (V.A.-G.); ibravo@ucm.es (I.B.-O.); iremm@ucm.es (I.T.M.-M.); 2Departamento de Farmacia Galénica y Tecnología Alimentaria, Facultad de Farmacia, Universidad Complutense de Madrid (UCM), IdISSC, 28040 Madrid, Spain; 3Charles Institute of Dermatology, School of Medicine, University College Dublin, D04 V1W8 Dublin 4, Ireland; sigen.a@ucd.ie (S.A.); wenxin.wang@ucd.ie (W.W.); 4Blafar Ltd., Belfield Innovation Park, University College Dublin, Belfield, D04 V1W8 Dublin 4, Ireland; hongyun.tai@blafar.com; 5Thematic Research Network in Ophthalmology (Oftared), Carlos III National Institute of Health, 28029 Madrid, Spain

**Keywords:** PLGA-PEG-PLGA, thermo-responsive hydrogel, micelles, neurodegenerative diseases, intravitreal drug delivery, oxidative stress, inflammation, ketorolac, dexamethasone

## Abstract

The present study aims to develop a thermo-responsive-injectable hydrogel (HyG) based on PLGA-PEG-PLGA (PLGA = poly-(DL-lactic acid co-glycolic acid); PEG = polyethylene glycol) to deliver neuroprotective agents to the retina over time. Two PLGA-PEG PLGA copolymers with different PEG:LA:GA ratios (1:1.54:23.1 and 1:2.25:22.5) for HyG-1 and HyG-2 development respectively were synthetized and characterized by different techniques (gel permeation chromatography (GPC), nuclear magnetic resonance (NMR), dynamic light scattering (DLS), critical micelle concentration (CMC), gelation and rheological behaviour). According to the physicochemical characterization, HyG-1 was selected for further studies and loaded with anti-inflammatory drugs: dexamethasone (0.2%), and ketorolac (0.5%), alone or in combination with the antioxidants idebenone (1 µM) and D-α-Tocopherol polyethylene glycol 1000 succinate (TPGS) (0.002%). In vitro drug release and cytotoxicity studies were performed for the active substances and hydrogels (loaded and drug-free). A cellular model based on oxidative stress was optimized for anti-inflammatory and antioxidant screening of the formulations by using retinal-pigmented epithelial cell line hTERT (RPE-1). The copolymer 1, used to prepare thermo-responsive HyG-1, showed low polydispersity (PDI = 1.22) and a strong gel behaviour at 25% (*w*/*v*) in an isotonic buffer solution close to the vitreous temperature (31–34 °C). Sustained release of dexamethasone and ketorolac was achieved between 47 and 62 days, depending on the composition. HyG-1 was well tolerated (84.5 ± 3.2%) in retinal cells, with values near 100% when the anti-inflammatory and antioxidant agents were included. The combination of idebenone and dexamethasone promoted high oxidative protection in the cells exposed to H_2_O_2,_ with viability values of 86.2 ± 14.7%. Ketorolac and dexamethasone-based formulations ameliorated the production of TNF-α, showing significant results (*p* ≤ 0.0001). The hydrogels developed in the present study entail a novel biodegradable tool to treat neurodegenerative processes of the retina overtime.

## 1. Introduction

Retinal diseases comprise one of the leading global causes of visual loss in the world. The proportion of the total visual impairment and blindness caused by age related macular degeneration (AMD), glaucoma and diabetic retinopathy (DR) involve neurodegenerative events and are currently greater than from infective causes [1]. AMD is the leading cause of irreversible vision loss in the world [2]. Although the pathophysiology is still undergoing research, several studies have focused on the retinal pigmented epithelium (RPE)/photoreceptor/Bruch’s membrane complex [3,4]. RPE is of vital importance in maintaining visual function of the photoreceptors present in the retina by supporting the metabolism and energetic requirements of photosensitive cells [3]. AMD can be subclassified into early, intermediate and late AMD. The main differences between these are the damage and progression in the fundus structures including neovascularization, atrophy and retinal degeneration [5]. If the disease progresses to late AMD, neovascularization can be present or absent, i.e., wet or exudative AMD, and dry or atrophic AMD. While wet AMD rapidly results in untreatable blindness, dry AMD is considered more as a chronic disease, that slowly evolves into irreversible vision loss [6,7]. Oxidative stress has also been related to AMD and its progression, being hypothesized that undesired metabolic debris that have not been properly eliminated become oxidized, promoting inflammation and cellular damage [8].

Glaucoma is another debilitating disease related to neurodegeneration. Among all the types of glaucoma, primary open angle (POAG) and closed angle (PACG) glaucoma are the most common [9]. Intraocular pressure (IOP) is the main risk factor of glaucoma and it is linked to the compaction of axonal dendrites and surrounding structures, leading to morphological and irreversible changes in the lamina cribosa [10]. These alterations together with the interruption of flow and nutrients to retinal ganglion cells cause cell death, apoptosis and loss of function. These alterations trigger an excessive production of reactive oxygen species (ROS), interacting with cell structures and up-regulating cell death [11]. There are other degenerative ocular diseases that inevitably cause visual disfunction, like diabetic retinopathy (DR) and retinitis pigmentosa (RP). DR is widely recognized as one of the main complications of diabetes mellitus and is classified into non-proliferative, as the early stage, and proliferative, as the advanced stage of DR. In diabetes mellitus, the extended periods of hyperglycemia promote retinal blood vessels dilation and micro-vascularization which has been associated with pericytes apoptosis and inflammation mechanisms [12,13]. Besides, massive ROS are produced, worsening inflammation and progression of DR [12]. Retinitis pigmentosa (RP) is also a major ocular debilitating disease that causes the loss of photoreceptors. RP pathophysiology has been widely studied and it seems that there is no single specific mechanism that may trigger the disease. Despite its complexity, RP has been linked to genetic mutations, from dominant/recessive to alterations in mitochondrial genetic information. As with some of the diseases mentioned above, in RP patients experience RPE degeneration progressing to cell death. Some of the genes affected in RP directly cause a dysregulation in cell apoptosis, inflammation mechanisms and cyto-protection against light-induced damage [14]. Furthermore, oxidative stress and ROS production worsen the effects of RP by triggering inflammation and apoptosis cascades of endothelial RPE cells and, ultimately, photoreceptors, leading to vision loss [15].

There are some therapies that have been developed in order to slow or revert blindness progression in these neurodegenerative diseases of the retina. Corticosteroids and particularly dexamethasone have been investigated, especially in combined therapies for their potent anti-inflammatory effects as well as their ability to stabilize the blood-retinal barrier and reduce exudative processes [16]. Furthermore, some non-steroidal anti-inflammatory drugs (NSAID), such as ketorolac or bromfenac, have been also used alone [17] or in combination with anti-VEGF therapies, in order to reduce retinal thickness caused by macular edema, counteract the inflammation processes in degenerative processes and avoid the inflammation caused by the intravitreal injection itself [18].

D-alpha-tocopherol polyethylene glycol succinate (TPGS) has been included in various drug delivery systems to enhance bioavailability and drug efficacy [19]. This vitamin-derivative possess the antioxidant properties of vitamin E and can be solubilized in an aqueous media [20]. Q10 coenzyme has also been studied for its capacity to protect retinal ganglion cells against damage and oxidative stress [21]. Recently, the partially soluble analogue of Q10 coenzyme known as idebenone has gained much interest as a neuroprotective therapy for Alzheimer, Parkinson and retinal degeneration. Idebenone has been described as possessing a strong antioxidant activity that is currently being intensively studied and could entail an interesting option in combined therapy for neuroprotection in retinal degenerative diseases [22].

Intravitreal administration is used to deliver drug therapies to the posterior segment of the eye [23]. However, difficulties related to the drug such as short residence time and high clearance, as well as the inflammation associated with frequently repeated injections and risk of visual impairment, are some of the common drawbacks of this kind of therapy. To overcome these limitations, investigations have been directed towards the design of intraocular drug delivery systems that are able to resolve some of the above-mentioned difficulties. This is the case with PLGA-PEG-PLGA triblock copolymers, which constitute a family of amphiphilic water-soluble polymers made out of poly-(DL-lactic acid co-glycolic acid) (PLGA) and polyethylene glycol (PEG) monomer units [24]. In aqueous solution, PLGA-PEG-PLGA copolymers are able to create micelle-like structures with a high hydrophobic core (PLGA) and a surrounding corona-like structure made of PEG tails [25]. Although the features can be specifically designed, normally at room temperature, the micelles are well separated and distributed, conferring to the solution a sol state. However, at higher temperatures, micelles increase their size and start to aggregate, entering the gel state and thus creating a thermo-responsive hydrogel response. It has been suggested that at higher temperatures a micelle destruction occurs due to PEG chain dehydration as well as polymer precipitation [26]. PLGA-PEG-PLGA copolymers could be a useful delivery platform to increase the residence time of active substances in the posterior segment of the eye, allowing the design of therapies avoiding frequent administration [27]. In the present study we assess the potential of a PLGA-PEG-PLGA triblock based hydrogel in a buffer isotonic solution for intravitreal injection. The hydrogel aims to deliver dexamethasone phosphate and ketorolac tris salt in combination with idebenone or TPGS to the posterior segment of the eye, as a novel neuroprotective therapy for retinal degenerative diseases.

## 2. Materials and Methods

### 2.1. Materials

Polyethylene glycol (PEG 1500), stannous 2-ethylhexanoate, dexamethasone phosphate, idebenone and D-α-Tocopherol polyethylene glycol 1000 succinate (TPGS) and ketorolac tris salt were purchased from Sigma-Aldrich (Leinster, Ireland). DL-Lactide and glycolide were purchased from Corbion^®^ (Gorinchem, The Netherlands). The dialysis membranes (Spectra-Por^®^ Float-A-Lyzer^®^ G2, 5 mL, MWCO 3.5–5 KDa) were from Sigma-Aldrich (Madrid, Spain). DMEM/F12 (Dulbecco’s Modified Eagle Medium/Nutrient Mixture F-12), Hank’s Balanced Salt Solution, H_2_O_2_ 50% and MTT were purchased from Sigma-Aldrich (Leinster, Ireland). FBS, L-glutamine and sodium bicarbonate were purchased from Invitrogen (Dublin, Ireland). TNF and IL1b kit for inflammation studies were purchased from Invitrogen (Dublin, Ireland).

### 2.2. Synthesis of the PLGA-PEG-PLGA Triblock Copolymers

Two different PLGA-PEG-PLGA (lactic acid (LA): glycolic acid (GA): polyethylene glycol (PEG)) triblock copolymers were synthesized according to the ring opening polymerization (ROP) method described previously by Yu L et al. [12] and Zentner et al. [13], with some modifications. In order to select the method with the most desirable behaviour for intravitreal injection, two different ratios of PEG:LA:GA were selected (1:1.54:23.1 and 1:2.25:22.5), establishing PEG 1500 as the initiator.

For the preparation of Copolymer 1, 15 g of PEG 1500 (0.01 mol) were dried in a two-necked flask under vacuum and stirred at 100 °C overnight. Under the protection of argon, 1.8 g of GA (0.015 mol) and 33.3 g of LA (0.231 mol) (GA:LA 1:15) were added to the mixture and heated with stirring at 130 °C for 1 h under reduced pressure. After all the monomers had melted, 0.070 g of stannous 2 ethyl hexanoate (0.2% *w*/*w* of the monomers) were added, and the reaction mixture was further heated at 150 °C for 8 h under argon atmosphere. Unreacted monomers were removed under vacuum for 60 min. Crude copolymer was dissolved in ice-cold water (5–8 °C). After complete dissolution, the copolymer solution was precipitated at 80 °C and water-soluble low molecular weight copolymer and unreacted monomers were eliminated by decanting the supernatant. Once the supernatant was decanted, crude copolymer was redissolved in ice-cold water. To obtain the final purified copolymer, the same process of heating, precipitation and decantation was repeated three times. Residual water was removed by freeze-drying, and the copolymer was stored at −30 °C. Copolymer 2 (GA:LA 1:10) was synthesized following the same procedure described above.

The chemical structure of the PLGA-PEG-PLGA triblock was determined by ^1^H-NMR. Spectra were recorded at 400 MHz on a Bruker Ascend spectrometer at 25 °C in deuterated chloroform (CDCl_3_). The weight average molecular weight was determined by gel permeation chromatography (GPC) using an Agilent 1260 series with a refractive index detector, a viscometer detector, a dual-angle light-scattering detector (LS 15° and LS 90°) and two series of Polargel-M (7.5 × 300 mm^2^) columns. The analysis was performed at 35 °C using dimethyl formamide as the eluent with a flow rate of 1 mL/min.

### 2.3. Preparation of the Combined Therapy Formulations

Ophthalmic formulations based on HyG-1 and HyG-2 developed with copolymer 1 and 2, respectively, were prepared by dissolving the corresponding copolymer (25% *w*/*v*) in an optimized bicarbonate buffer composed of NaHCO_3_ 0.095M, Na_2_CO_3_ 0.005M (buffer pH = 8.81). After addition of the corresponding active compounds, trehalose was also included at 3.25% or 2.33% in the preparation of dexamethasone or ketorolac formulations, in order to adjust the final pH (≈ 7–7.4) and isotonicity (≈300 mOsm/L). Furthermore, idebenone and TPGS were also combined with dexamethasone or ketorolac as described in Table 1.

### 2.4. Characterization of Polymers and Final Formulations

To study the thermo-responsive behaviour of the hydrogel in the different aqueous media, different polymer solutions at given concentrations of 15%, 20% and 25% (*w*/*v*) were prepared in several vehicles such as distilled water, PBS, Hanks, NaCl 0.9% and the above-mentioned isotonic bicarbonate buffer including trehalose. The sol (flow)—gel (nonflow) transition of the copolymers in the different aqueous vehicles was determined using the inverting test method in a 10 mL vial by increasing the temperature by1 °C every minute from a starting point of 25 °C to 40 °C.

The viscosity of PLGA-PEG-PLGA hydrogels (HyG-1 and HyG-2) prepared in water and the other isotonic solutions were studied using a Discovery HR-hybrid Rheometer (New Castle, DE, USA) with a parallel plate system (8- or 20-mm diameter). To this, shear rates increasing from 0 to 1000 s^−1^ in 20 steps were used. Viscosity was measured when the steady state was reached. The determination was performed within the temperature range 25 °C–45 °C at a heating rate of 0.6 °C/min at 10 rad/s. The rheology (G’ module vs. time) of final formulations (Table 1) were studied under the same conditions.

The critical micelle concentration (CMC) in the bicarbonate buffer/trehalose was measured with a tensiometer (K-11, Kruss) by using the Wilhelmy plate method. Before sampling analysis, the tensiometer was calibrated with MilliQ water (72.0 ± 1 mN/m) and the single bicarbonate buffer was measured (67.3 ± 0.5 mN/m). In order to obtain the desired curve, thirteen different concentrations of the hydrogels were used ranging from 0.010 mg/mL to 5 mg/mL. The equilibration time was set to 3 min and every concentration was measured in triplicate. In addition, the average size and size distribution of the micelles were determined using a laser scattering spectrophotometer (Autosizer 4700, Malvern), with a vertically polarized incident beam at 532 nm supplied by an argon ion laser. The scattering angle of measurements was set at 90 °C.

### 2.5. In Vitro Release Studies

In order to evaluate the drug release profile of HyG-1 based formulations, in vitro studies were conducted as depicted in Figure 1. Firstly, 5 mL of each formulation described in Section 2.3 were introduced in a pre-activated dialysis membrane (Spectra-Por^®^ Float-A-Lyzer^®^ G2, 5 mL, MWCO 3.5–5 KDa) and put in 50 mL closed falcon tubes filled with 40 mL of PBS at 37 °C. The tubes were introduced into a water bath at the same temperature. The release media was extracted and replaced with pre-warm (37 °C) fresh media at preset times maintaining sink conditions. During the first day, intervals of 1 h, 4 h, 8 h 12 h and 24 h were used. In the next 10 days, the media was replaced every 24 h, then twice a week until the end of study. Each formulation was developed and assessed in triplicate.

Drug release of dexamethasone phosphate and ketorolac tris salt was quantified by HPLC as previously described by H. AlAani et al. with some modifications [28]. The study was performed with a Waters Alliance 2695 separation module equipped with Waters photodiode array 2996 (Waters corp, Barcelona, Spain). Dexamethasone was determined at a wavelength of 240.5 nm and ketorolac at 317 nm. For both determinations, the mobile phase was composed of 0.05 M potassium dihydrogen phosphate in water (adjusted to pH 4 with formic acid) and acetonitrile (70:30) at a flow rate of 1 mL/min. The column used for the chromatographic separation was a Hypersil silica column (250 × 4 mm 5 µm particle size) and Empower3 software (Waters, Barcelona, Spain) was employed for data acquisition and processing.

The release kinetics of dexamethasone and ketorolac from the respective hydrogel formulations were analyzed according to Korsmeyer-Peppas model (Equation (1)). This semi-empirical model offers information about the mechanism involved in drug release. Although it was initially proposed for describing release behavior from solid monolithic systems such as tablets, it has subsequently been widely used to successfully describe the drug release mechanism from other more sophisticated drug delivery systems such as microparticles [29], liposomes [30] or semisolid systems such as hydrogels [31,32].
(1)$$\frac{M_{t}}{M_{\infty }}=K\cdot t^{n}$$
where *t* is the release time, *M_t_* is the amount of drug delivered at time *t*, *M*_∞_ is the total amount of drug delivered, *K* is a kinetic constant, and *n* the diffusional exponent that indicates the drug release mechanism. *M_t_/M*_∞_ values lower than 0.6 were used for fitting, and sink conditions were maintained throughout the dissolution assay. According to the authors, a value of *n* equals 0.5 describes a Fickian diffusion of the drug through the polymeric matrix (Case I), and values between 0.5–1.0 describe an “anomalous (non-Fickian) transport”, in which not only diffusion but other mechanisms (i.e., a mixed diffusion and chain relaxation mechanisms) are involved in the control of the drug release. When *n* takes a value of 1.0, this implies a Case II transport, describing a predominant influence of polymer relaxation on the guest molecules’ movement within the matrix [33]. These values are valid for slab geometry. In fact, they have been modified, for example in the case of cylinders or spherical systems [34]. However, for hydrogel evaluation the initial values are commonly assumed [31,32], and they will be followed in the present study.

The similarity factor (*f*_2_) (Equation (2)) was chosen to compare the dissolution profiles. This is a logarithmic transformation of the sum-squared error of differences between the test and reference products over all time points:(2)$$f_{2}=50\cdot \log\left[\frac{1}{\sqrt{1+\frac{\sum_{t=t}^{t=n}{\left(R_{t}-T_{t}\right)}^{2}}{n}}}\cdot 100\right]$$
where *n* is the number of experimental points in the in vitro dissolution assay, *R_t_* and *T_t_* are the mean percentages of dissolved drug from the reference and test formulations, respectively, at each time point *t*. No more than one sampling time point after 85% dissolution was considered. When values of *f*_2_ are between 50 and 100, it can be ensured that the difference between the two release profiles under study is lower than 10% [35].

### 2.6. Cell Cultures

A retinal-pigmented epithelial cell line hTERT (RPE-1) kindly donated by the UCD (University College of Dublin) Conway Institute was used to evaluate cytotoxicity of the formulations, screen anti-inflammatory properties and develop an oxidative stress model. Cells were cultured at 37 °C (5% CO_2_) in Dulbecco’s Modified Eagle Medium/Nutrient Mixture F-12 (DMEM/F-12) supplemented with FBS 10%, L-Glutamine 1% and 3.5% of sodium bicarbonate 7.5%. RPE-1 cells were split at 80% of confluence, 1–2 times per week and the number of passages at the moment of the experiments was 10.

### 2.7. Cell Viability

In vitro cytotoxicity was assessed using mitochondrial-dependent reduction of the tetrazolium salt, 3-(4-5-dimethylthiazol-2-yl)-2,5-diphenyltetrazolium bromide (MTT) to formazan. Cytotoxicity studies were performed with different solutions of ketorolac, dexamethasone, HyG-1 and HyG-2, as well as the final developed formulations. Briefly, 50 μL of the tested materials were added to each well (96 well plates) and incubated at 37 °C 5% CO_2_ for 1 h to ensure proper gelation. Then, in order to evaluate toxicity by contact, cells were seeded at a final number of 7000 cells/well, on top of each polymer/formulation. As positive control, a solution of DMSO (dimethyl sulfoxide) 15% was used. Cells were grown for 24 h and the supernatant was removed cautiously in order to avoid the aspiration of the polymers in gel state. Finally, the wells were filled with a mixture of MTT 5 mg/L and DMEM/F12 with FBS 1%, L-Glutamine 1% and 3.5% of sodium bicarbonate 7.5% in a 1:6 ratio (final concentration of MTT = 0.83 mg/mL). After the plates were incubated overnight, DMSO was added to each well in order to solubilize the formazan crystals. The plate was measured at 550 nm in the spectrophotometer.

### 2.8. Protective Activity to Oxidation in Cell Cultures

The protective oxidation activity of final formulations prepared with HyG-1 was performed via MTT assay in cells exposed to H_2_O_2_. In order to optimize the H_2_O_2_ concentration to be used as oxidative stress positive control, different solutions were prepared from a H_2_O_2_ 50% concentrated standard (200, 150, 100, 50, and 25 μM).

To evaluate the protective antioxidant activity of the hydrogel-based formulations, each well was filled with 50 μL of the formulations and incubated at 37 °C (5% CO_2_) for 1 h to ensure proper gelation. Then, cells were seeded (7000 cells/well) and incubated overnight. Briefly, the supernatants were discarded, and each well was filled with a mixture of H_2_O_2_ in NaCl 0.9% (100 μL) and supplemented DMEM/F12 (100 μL) (1:1 ratio) to achieve a final concentration of 150 μM H_2_O_2_. After that, the cells were incubated for 24 h. The supernatants were removed and the MTT was added following an overnight incubation. Finally, DMSO was added and the plates were measured at 550 nm as previously described.

### 2.9. In Vitro Anti-Inflammatory Activity in Response to LPS Stimulation

Anti-inflammatory activity was evaluated in RPE-1 cells by human ELISA kit anti TNF alpha. Insert trans-wells (0.4 μm pore size) were filled with 300 μL of each hydrogel formulation in a 24 well plate and 2.5 × 10^5^ cells were seeded on the bottom of each well. Plates were incubated for 24 h and the media was then removed. Then 5 μg/mL of LPS in supplemented DMEM/D12 with FBS 1% were added to each well and the cells were incubated for another 24 h. Supernatants were centrifuged at 850× *g* and measured by ELISA anti-TNF alpha as stated in the protocol of the commercial kit.

### 2.10. Statistics

The experiments were performed in triplicate (*n* = 3). Data are represented as mean ± standard deviation (SD). One-way ANOVA with Dunnett’s test was used to determine significance (*p* ≤ 0.05, significant; *p* ≤ 0.01, very significant or *p* ≤ 0.001, highly significant) using GraphPad Prism Version 9 (GrapPad software, San Diego, CA, USA).

## 3. Results

### 3.1. Synthesis and Characterization of Crude Copolymers

The molecular weight of the PLGA-PEG-PLGA copolymer was selected based on the needs of the ophthalmic application, in order to achieve a proper and quick gelation of the polymer solution at intravitreal temperature (31–34 °C) [26]. The main structure of the PLGA-PEG-PLGA triblock copolymers 1 and 2 was confirmed by hydrogen nuclear magnetic resonance (^1^H-NMR) (Figure 2). ^1^H-NMR results showed that experimental ratios of copolymer 1 were closer to the theoretical and more homogeneous than those of copolymer 2. Moreover, chemical shifts from PLGA-PEG-PLGA were identified showing the typical structure of triblock copolymers. Briefly, CH groups of lactic acid (LA) were found at 5.20 ppm (a), CH_2_ of glycolic acid (GA) at 4.75 ppm (b), CH_2_ pertaining both signals to PEG at 4.30 ppm (c) and 3.65 ppm (d) and CH_3_ of LA at 1.55 ppm (e). Experimental and theoretical ratios of both copolymers are calculated in Table 2 according to other authors [36].

The GPC results for copolymer 1 and 2 showed that the polydispersity index (PDI) values were between 1.22 and 1.36, respectively, resulting in unimodal molecular weight distributions. Number average molecular weight, weight average molecular weight and PDI are shown in Table 2. Furthermore, Figure 3 shows the molecular weight distribution for both copolymers (1 and 2). As can be appreciated, the distribution of areas appears unimodal.

### 3.2. Physicochemical Characterization

#### 3.2.1. Sol-Gel Transition Temperature

The transition temperature from sol to gel state is shown in the phase diagram of Figure 4. Copolymer 2 showed a sol gel transition at earlier temperatures (from 27 °C to 29 °C) than copolymer 1 (from 31 °C to 33 °C). Besides, regarding the precipitation temperature, copolymer 1 exhibited a higher stability at temperatures between 33–39 °C with a precipitation window starting at 39 °C. On the contrary, copolymer 2 displayed a precipitation window between 34–39 °C. In both cases, the transition temperature was found to be dependent on the concentration and reversible. When cooled, the copolymers exhibited hysteresis by reversing from gel to sol more slowly than from sol to gel. Both copolymers showed a sol state fully transparent, and a gel state transparent at the beginning. While the temperature was increased, hydrogels slowly became opaque until a precipitate state was shown. At this point an irreversible transition to sol and a phase separation was exhibited. Copolymer 1 demonstrated a wider range of transparence at gel state and higher stability than copolymer 2 at physiological temperatures of the posterior segment of the eye (32–37 °C).

#### 3.2.2. Rheometry and Viscosity Analysis

Rheological behaviour of copolymer 1 and 2 at different concentrations in water, bicarbonate buffer and the final loaded formulations were studied. Both copolymers 1 and 2 exhibited sol behaviour with low viscosities at room temperature (25 °C) demonstrating their suitability for injectable applications (Figure 5). However, copolymer 2 showed lower viscosity intensity than copolymer 1, lower gelation temperatures, and more instability at 37 °C. On the contrary, copolymer 1 exhibited stronger gelation, more stability at physiological temperatures and wider gelation spectrum than copolymer 2. Besides, copolymer 1 in the injectable bicarbonate buffer appeared more stable with a more sustained gelation overtime.

Final formulations with HyG-1 were prepared by dissolving copolymer 1 in the selected bicarbonate buffer followed by addition of the active substances. Their rheological behavior is shown in Figure 6. Both formulations (with dexamethasone phosphate or ketorolac tris salt) showed sol state properties at room temperature, starting the sol-gel transition around 30 °C for both groups, and reaching the highest point at 34 °C. Dexamethasone phosphate-based formulations exhibited a wider range of variation between temperatures, the one that combined dexamethasone (DX) and idebenone 1 μM being the formulation with the strongest gelling behavior. For both sets of groups the gel started to tear apart when it reached 39–40 °C, decreasing the gel state dramatically.

#### 3.2.3. Size of Micelles

The size of the micelles of HyG-1 in the selected buffer and HyG-1 developed formulations was measured by dynamic light scattering (DLS) with a Malvern laser spectrometer (Figure 6). The average size for selected copolymer 1 in the bicarbonate buffer solution at 25% (*w*/*v*) without drugs was 24.12 ± 0.17 nm. All the formulations with dexamethasone phosphate and ketorolac tris salt are shown in Figure 7. Among the different formulations developed, micelles from the formulation containing DX 0.2% + TPGS 0.02% were found to have the smallest size (23.03 ± 0.23 nm) while formulations with KT 0.5% presented the highest values (34.80 ± 0.81 nm). All presented similar PDI values with unimodal distributions (Figure 7).

#### 3.2.4. Critical Micelle Concentration (CMC)

CMC measurements were used to confirm the critical concentration at which the polymer formed stable micelles, which will then assemble into a group of micelles to create the thermo-responsive gel. The CMC for the selected copolymer 1 was 0.13 mg/mL at 25 °C. This value was calculated by extrapolation of the two lines that create the surface tension/concentration curve, when crossing each other (Figure 8). Also, critical micelle concentration (CMC) is a useful way to confirm the concentration at which PLGA-PEG-PLGA chains start to aggregate and create micelles. Our selected copolymer 1 with a CMC of 0.13 mg/mL is in the range described in a previous published study (0.1–0.3 mg/mL) [26].

### 3.3. In Vitro Release Studies

The cumulative in vitro release profile of dexamethasone or ketorolac alone and in combination with TPGS 0.02% or idebenone 1 µM from hydrogels is presented in Figure 9 and Figure 10. Regarding dexamethasone formulations, all three showed a similar initial 24 h burst with values of 8.60 ± 0.11%, 8.94 ± 0.53% and 10.02 ± 0.86% for DX 0.2% (A), DX 0.2% + idebenone 1 µM (B) and DX 0.2% + TPGS 0.02% (C). respectively. A similar biphasic release profile was observed subsequently in the three cases. In the first phase, a sustained slow release was observed until day 51, with a release of 54.09 ± 0.87% for formulation A, of 73.25 ± 0.87% for formulation B and of 73.30 ± 0.88% for formulation C (Figure 9). After that, an increment in release rate from day 51 to the end of the assay at day 62 was also reported for the three hydrogel formulations. According to the Korsmeyer-Peppas model (Table 3), while the hydrogel formulation that contained only dexamethasone showed a predominant diffusion mechanism, with *n* values slightly lower than 0.5 [31,37], whether both TPGS or Idebenone were included in the formulation, an anomalous transport (*n* values between 0.5 and 1) was observed. This indicated that not only diffusion but also polymer chain related events, for example hydrogel erosion [32], have to be considered in these systems. The high values observed for the coefficient of determination of the fitting obtained for the three profiles confirm the applicability of the release model to describe the mechanism behavior.

The similarity factor calculated with formulation A as reference (only loaded with dexamethasone) demonstrated that the inclusion of both TPGS or Idebenone significantly modified the drug release profile, considering they are not similar (*f*_2_ values of 44 for Formulation C and of 41 for formulation D)

Ketorolac-based formulations exhibited higher initial 24 h burst in comparison with dexamethasone-loaded hydrogels. Values of 16.89 ± 1.86%, 21.87 ± 0.70% and 15.40 ± 0.06% were observed for KT 0.5% (D), KT 0.5% + Idebenone 1 µM (E) and KT 0.5% + TPGS 0.02% (F), respectively (Figure 10). After that, a single release phase was observed for the three hydrogel formulations until the complete release of ketorolac, that occurred at day 47 for formulation E and at day 53 for formulations D and F. The Korsmeyer-Peppas model was suitable to describe the release mechanism involved (Table 3). According to the exponent *n* value, in the three cases an anomalous transport of the drug was observed, meaning that the release mechanism involved might be a mixture of drug diffusion and other polymer chain related events.

Calculations of f_2_ considering formulation D as reference (only loaded with ketorolac), demonstrated that formulations E and F were similar to the reference, with values of 52 for the formulation including Idebenone and 90 for the formulation including TPGS.

### 3.4. Toxicity Assessment of PLGA-PEG-PLGA Copolymers and Formulations

Cellular toxicity in RPE-1 cells of the single active ingredients (dexamethasone phosphate and ketorolac tris salt) to be incorporated in the hydrogel were assessed in order to determine which concentration would be the most suitable in terms of tolerability (Figure 11).

Dexamethasone phosphate demonstrated high cell viability values at all concentrations evaluated, with values higher than 80% in all cases. Conversely, ketorolac tris salt showed a correlation between concentration and toxic effects, obtaining acceptable tolerance values (higher than 80%) only for concentrations lower than 0.5%.

The toxicity of copolymer 1 and 2 was evaluated in order to select the most appropriate example with the best above-mentioned characteristics. Copolymer 1 resulted in higher tolerance values than copolymer 2 at all concentrations studied, particularly at the concentration of more interest (25%) (Figure 12).

After determining cell viability of the active ingredients and polymers separately, the tolerance of the final formulations was also tested and shown in Figure 13. At 24 h, cell viabilities for dexamethasone-based formulations (A, B and C), were 133.10 ± 13%, 135.4 ± 8.44% and 129.4 ± 10.75%, respectively. For ketorolac-based formulations (KT 0.5%, KT 0.5% + Idebenone 1 µM and KT 0.5% + TPGS 0.02%) cell viabilities were 102.7 ± 6.49%, 107.0 ± 11.21% and 103.4 ± 15.71%, respectively.

### 3.5. Evaluation of Protective Properties in an Oxidative Stress Model

An oxidative stress model based on the MTT viability assay was developed in order to assess the protection of the formulations developed under an oxidative environment, simulating the conditions given in retinal degenerative processes. Different concentrations of H_2_O_2_ (200, 150, 100, 50 and 25 µM) were recreated as explained in Section 2.8. The different values of cell survival under H_2_O_2_ exposure are shown in Figure 14. According to the results, 150 µM H_2_O_2_ was selected to further evaluate the protection of final formulations against oxidative environments (25.91 ± 5.48 % cell survival).

As mentioned previously, cells were exposed to the protective final formulations and to the oxidative stress conditions selected from a previous optimization process (150 µM) (Figure 14). According to the results, all the formulations exhibited high levels of protection to cell death caused by H_2_O_2_ showing high statistically significant results (*p* ≤ 0.0001) in comparison with the positive control. Among all examples, the combination of DX 0.2% + Idebenone 1 µM and KT 0.5% alone were able to protect cells from oxidation, showing the highest viability values (86.24 ± 14.68% and 84.34 ± 8.65%, respectively) (Figure 15).

### 3.6. In Vitro Anti-Inflammatory Activity in Response to LPS Stimulation

The effect of LPS in RPE-1 activates the stimulation of TNFα production. By exposing the cells to LPS and the formulations in the trans-wells for 24 h, a clear inhibitory effect of TNFα can be seen (Figure 16). As a reference, the basal TNFα production was 4.79 ± 2.41 pg/mL and the positive control (cells exposed to 5 µg/mL of LPS without anti-inflammatory agents) showed a TNFα concentration of 34.69 ± 3.07 pg/mL. Ketorolac-based formulations were demonstrated to have the highest anti-inflammatory effect with TNFα values of 12.79 ± 2.24 pg/mL, 12.83 ± 4.64 pg/mL and 15.53 ± 5.57 pg/mL for KT 0.5%, KT 0.5% + Idebenone 1 µM and KT 0.5% + TPGS 0.02% each, with very high statistically significance (*p* ≤ 0.0001) (Figure 16). Dexamethasone and its combinations were also able to decrease the TNFα values to a lesser extent than KT with values between 21.64 and 24.86 pg/mL, being statistically significant in comparison with the positive control for inflammation (*p* < 0.01).

## 4. Discussion

Thermo-responsive hydrogels have been suggested as a novel tailored tool to successful administer injectable therapies [26,38]. In this work, two PLGA-PEG-PLGA based copolymers have been developed and characterized as suitable platforms for injectable formulations that deliver neuroprotective agents with anti-inflammatory and antioxidant activity to the posterior segment of the eye. In previous studies, some authors have studied the potential of PLGA-PEG-PLGA hydrogels at 10% *w*/*v* with glycolic to lactide ratios (1:3) and PEG 1000 to deliver NSAIDs, such as naltrexone in a sustained manner, therefore avoiding repeated daily administrations and achieving more effective drug concentrations [24]. In ocular applications, Yuan Gao et al. studied the development of a dexamethasone acetate (0.1%) loaded PLGA-PEG-PLGA hydrogel at 20% *w*/*v* as an alternative to effective dexamethasone eye drops and also suggested the possibility of using PLGA-PEG-PLGA to develop therapies for the posterior segment of the eye [36]. Thermo-responsive nanocomposite-based hydrogel made out of cefuroxime nano-emulsions and solid lipid nanoparticles dispersed in Pluronic^®^F127 have also been developed for the treatment of endophthalmitis [39]. To our knowledge, there have not been previous studies that propose and develop a complete optimized and fully characterized thermo-responsive formulation, biocompatible and biodegradable, including a combination of various neuroprotective agents to be administered in the posterior segment of the eye.

The basic structure of PLGA-PEG-PLGA copolymer 1 and 2 developed in this work presented the same groups and chemical shifts and were similar to other PLGA-PEG-PLGA polymers studied by other authors according to the ^1^H-NMR spectrum [24,26,36]. Besides, GPC peaks and PDI for both copolymers 1 and 2 (1.22 and 1.36) showed low polydispersity, uniformity and a unimodal distribution. One important aspect in the development of preparations for intravitreal administration is the resultant pH, in order to avoid any harmful effects due to pH deviation from physiological values [40]. Some studies have previously described the use of different aqueous vehicles like acetate buffer [41], water [42] or sodium chloride (NaCl) to prepare PLGA-PEG-PLGA based hydrogel formulations. However, in the present work low pH values (around 2–3) were rendered when using these aqueous vehicles in development of PLGA-PEG-PLGA hydrogels. To avoid this, both copolymers were dissolved in a bicarbonate buffer that counteracted the acidic nature of the polymer and equilibrated isotonicity with trehalose, so the formulations could be well tolerated and effective.

Copolymer 1 exhibited a crystalline sol behaviour at room temperatures (20–27 °C) which makes it ideal for injections. When reaching 32 °C, the dissolved copolymer 1 at a concentration of 25% (*w*/*v*) creates a transparent hydrogel, which is maintained for at least two to three weeks, depending on the substances entrapped. In addition, the gel modifies its structure by absorbing water and evolving into a high viscous semi-gel state. Furthermore, at this concentration the precipitation temperature starts at 39 °C, that being considered as a stable system for intravitreal delivery. On the contrary, lower concentrations (15% and 20% *w*/*v*) of hydrogel solutions started to gel at higher temperatures (Figure 4). Furthermore, in these conditions the gel was not strong enough and lasted only for a few days, and precipitation underwent was close to physiological temperatures (36–37 °C). Regarding copolymer 2, 15%, and 20% (*w*/*v*) solutions also developed weaker gels, although fully transparent. However, all the tested concentrations of HyG-2 (15%, 20%, and 25%) presented sol-gel transition temperatures very close to room temperature, showing some viscous behaviour with normal room temperature variation. These features made it difficult to work with and to check its suitability for injection.

The rheological behaviour of copolymer 1 based hydrogels confirmed the strength mentioned previously. They also show a maximum gel point at around 34–36 °C just in the range of physiological temperatures of vitreous humor. At higher temperatures the gel starts to tear down, which confirms the precipitation process. The optimized bicarbonate buffer increases the gel strength and more sustained hydrogels over time. These findings, and others that will be further discussed, envisaged that copolymer 1 at 25% (*w*/*v*) presented more suitable characteristics to develop an optimal formulation for intravitreal injections. Another important feature is the size distribution of the micelles that create the hydrogel, since their aggregation and therefore gelation could differ. Some studies have shown that DLS is the most common technique to study size distribution, and that PLGA-PEG-PLGA micelle sizes of around 21–31 nm, such as those obtained in the present work, are within common scope, although aggregation could occur and therefore increase size [26]. In the present study, final formulations prepared with copolymer 1 showed different micelle sizes depending on the substances included. Dexamethasone-HyG-1 based formulations (A, B, C) showed smaller micelle sizes than ketorolac-based formulations (D, E, F). According to the sizes presented, we hypothesize that dexamethasone interacts more closely with PLGA chains that ketorolac since dexamethasone phosphate presents almost ten times less water solubility (1.52 mg/mL) than ketorolac tris salt (15 mg/mL) [43,44]. Additionally, formulations C and F containing TPGS presented smaller size distributions than the rest of their group. These findings could suggest that the presence of TPGS at 0.02% could interact in some way with micelles, stabilizing them thanks to its amphiphilic nature, promoting lower sizes and PDI values [45]. Polydispersity index values were relatively high in all cases. This behavior has also been observed by other authors and explained due to the high molecular weights and some level of branching that are sometimes observed in triblock-copolymers [45]. Additionally, some micelle aggregation due to the relatively high viscosity of the sample cannot be discarded.

One critical characteristic that is worth mentioning is the in vitro release profile. Sustained release that ensures a certain level of the drug overtime entails a great advance in order to reduce the number injections needed, increase effectiveness and patient’s quality of life. In this sense, all HyG-1 based formulations proposed in this work showed sustained release of the loaded active compounds. The initial bursts during the first 24 h were well controlled in both dexamethasone and ketorolac-based hydrogels, although ketorolac presented a slightly higher burst than dexamethasone-based examples, probably due to its higher hydrophilicity. These initial bursts were followed by a sustained release, showing the utility of HyG-1 based formulations as sustained drug delivery systems. In the case of ketorolac-based hydrogels, the systems were able to release the drug for 47–53 days depending on the combinations, this time being extended for dexamethasone hydrogels to 62 days. A sustained in vitro release of drugs was also observed for a thermo-responsive hydrogel based on PEG-PCL (40% at 14 days) [46]. Comparison of dexamethasone-based hydrogels’ drug release profiles demonstrated that the inclusion of both TPGS and idebenone dramatically increased the release of dexamethasone, considering they are “not similar” according to f_2_ values. We hypothesize that, according to the partition coefficient, idebenone (sparingly soluble in aqueous buffers with a logP of 4.3) [47] interacts with HyG-1 PLGA-PEG-PLGA micelles and shifts dexamethasone phosphate (logP value of 1.9) from interacting with micelles, therefore increasing the amount of free drug that is released [43]. TPGS presents higher hydrophilicity than idebenone and dexamethasone, respectively, with a solubility in water of 20% *w*/*v* [48]. This could also affect the capacity of dexamethasone to interact with HyG-1 micelles and increase its release. It has been previously reported that TPGS is an absorption enhancer and possess emulsification properties. Mustafa et al. developed PLGA nanoparticles loaded with kanamycin and emulsified with TPGS. They demonstrated faster release profiles with TPGS based nanoparticles, showing the ability of TPGS to interact with PLGA chains [49]. We can ask whether TPGS emulsification properties could have enhanced dexamethasone release. This modulation in the release could be due to the ability of PEG chains present in TPGS to interact with several PEG chains of PLGA-PEG-PLGA micelles and shift dexamethasone to the aqueous phase, therefore increasing its release. Conversely, dexamethasone formulations without any additional substances seemed to decrease its release ratio until the late stages, where they seemed to undergo the rupture of micelles and hydrogel, thus releasing the rest of the drug (Figure 9). The addition of either TPGS or idebenone in dexamethasone based-hydrogels also seemed to influence the mechanisms involved in the drug release until 60% of total dose, shifting from a simple diffusion of the drug through the hydrogel network in the case of dexamethasone alone to a mixture of that phenomenon with others related with micelle movement/rearrangement or hydrogel erosion [32]. Regarding ketorolac in vitro release, there are no studies that combine ketorolac with idebenone in a thermo-responsive hydrogel, so little is known about their interactions. However, and according to what has been previously claimed for dexamethasone, it is plausible that as well occurring with dexamethasone, idebenone shifts the active ingredient to the aqueous phase and increases its release. Nonetheless, taking into account that ketorolac tris salt is much more soluble than dexamethasone phosphate, this would not be affected by the addition of TPGS. Furthermore, TPGS is not included in high concentrations (0.02%) and the ketorolac tris salt release rate is elevated. This could be the reason why single ketorolac and its combination with TPGS would not be affected as much as in the case of dexamethasone. It is also worth mentioning that between the three in vitro profiles of ketorolac-based formulations, differences in the release profile are more subtle than for dexamethasone. In fact, this difference is lower than 10%, according to f_2_ values, and in the three cases a similar release mechanism based on diffusion plus polymeric chain rearrangement/erosion can be described, according to the Korsmeyer-Peppas model. As mentioned earlier, in part this would be due to the solubility of both active ingredients, ketorolac tris salt being almost eight-fold more soluble in water than dexamethasone phosphate [43,50].

For cell in vitro tolerability studies, we took as an initial step in the preparation of the copolymer 1 based formulations the screening of safe concentrations of dexamethasone phosphate (0.2%) and ketorolac tris salt (0.5%). Moreover, the toxicity of HyG-1 and HyG-2 prepared at different concentrations (15%, 20%, and 25% *w*/*v*) was also tested after 24 h contact in order to ensure that the hydrogels were well tolerated in retinal cells. To test the toxicity of hydrogels, a new protocol was developed, since the gels were so strong that they did not allow the MTT to penetrate into the cells. Therefore, proper gelation onto the plates needed to be established before seeding the cells on top of the gels. This entails a novel way of testing toxicity by direct contact with these biomaterials since, until now, these types of hydrogels have generally been tested by indirect toxicity, by exposing cells to degradation products of the hydrogels [51]. According to the performed toxicity assay, HyG-1 is much better tolerated than HyG-2, the concentration of 15% being the only one showing viability values superior to 80%, a limit commonly accepted as well tolerated according to the criteria for ocular drug delivery testing in cell cultures [52]. These toxicity studies, together with the physicochemical data (low sol-gel-precipitation transition temperature, unstable rheological behaviour and higher polydispersity) observed for HyG-2, allowed us to discard it for further steps in the experimental work. On the contrary, HyG-1 has the criteria for being well tolerated, creating stable and transparent gels at the physiological temperatures of the vitreous, exhibiting high precipitation temperatures which avoid its degradation, and possessing good injectability at room temperatures. Among the three concentrations of copolymer 1, 25% (*w*/*v*) was the one selected as presenting the best characteristics in terms of sol-gel properties, long-term in vitro drug release and good tolerability. HyG-1 formulations at 25%, including dexamethasone or ketorolac and their combinations were also tested in terms of cell tolerance. Among these, dexamethasone-based formulations exhibited exceptionally good tolerance in RPE-1 (with values even higher than 100%) after 24 h exposure. According to some studies, these findings could be linked to the ability of dexamethasone to stimulate cell proliferation together with the low toxicity of drug combination and polymers [53,54,55]. In addition, ketorolac-based formulations also showed viability values close to 100%.

It is well known that oxidative stress plays a very important role in the development of neurodegenerative diseases of the retina [56]. Recently, Masuda et al. highlighted the close relation between oxidative stress and neurodegenerative retinal diseases. They gathered together studies where inflammation was a triggering factor of ROS, starting oxidative stress and an inflammatory cascade. Among the cytokines studied, TNF-α entails a pivotal point of the inflammation cascade [57]. Besides, many neurodegenerative processes of the retina such as AMD or diabetic retinopathy start or progress with lack of retinal perfusion leading in many cases to ischemia [58]. Some studies have already established a solid relation between ischemic processes and ROS generation due to the neovascularization process, hyperglycemic states leading to lipid peroxidation, or finally hypoxia [58]. The massive production of ROS and inflammation eventually leads to cell apoptosis, loss of functionality of retinal pigmented epithelium, photoreceptor loss and irreversible blindness. In this study, we have used a method to evaluate the protection of different formulations by generating cell death in RPE-1 cells produced by the oxidative stress of H_2_O_2_. Since H_2_O_2_ is one of the main radicals produced under oxidative stress in many retinal degenerative processes [59], we selected it to screen potential neuroprotective agents. As mentioned previously, different concentrations of H_2_O_2_ were tested in order to select one that allowed us to evaluate protective effects. Among them, 150 μM was chosen as the oxidative stress stimulus that showed cell viability values of 25.91 ± 5.48%. This model allowed the evaluation of whether a developed formulation can preserve cell death provoked by oxidative stress. According to the studies performed, all the developed formulations showed statistically significant results (*p* < 0.0001) resulting in high protective formulations. Among them, formulation B (DX 0.2% + Idebenone 1 µM) and formulation D (KT 0.5%) showed the highest survival ratios (86.24 ± 14.68% and 84.34 ± 8.65%, respectively), more than four-fold higher than the cells directly exposed to H_2_O_2_ without protection, used as positive control. With regards to these results, it has been reported that dexamethasone is able to suppress some important signaling pathways such as p38 MAPK and NF-κB, both involved in retinal inflammation. Besides, they also reported that dexamethasone protected against ROS and increased survivability of RPE cells and photoreceptors [60]. In its turn, idebenone is an analogue of Q10 coenzyme that has been demonstrated to act as a very effective electron carrier, therefore trapping ROS generated by the excess of H_2_O_2_, and avoiding interaction with cell membranes [61]. In the case of Ketorolac, its antioxidant effect could also be attributed to its capacity to inhibit cyclooxygenase (COX). In fact, some authors have previously reported that COX triggers ROS production which feeds the inflammatory cycle. Besides this, the activity of ketorolac in protecting cells from ROS has been well studied [62]. Besides, TPGS in combination with dexamethasone (formulation B), also serves to preserve cell death provoked by oxidative stress (68.48 ± 6.81% TPGS appears as a very attractive hydrophilic alternative to vitamin E since it can be fully dissolved in aqueous buffers. Moreover, TPGS has been described as protecting retinas from ischemia-reperfusion injuries as well as against free radicals produced by lipid peroxidation [63]. On the other hand, ketorolac is a very potent NSAID that has been proposed to decrease inflammation in retinal degenerative diseases such as AMD [64] and appears to enhance its activity in combination with idebenone (71.54 ± 5.89% cell survival). It is worth noting that, although some studies have pointed out that dexamethasone might induce oxidative stress [65], our findings showing potential protective properties are supported by other authors that encourage the use of intravitreal dexamethasone for retinal degenerative diseases [66]. Regarding inflammation, there is no doubt that all the formulations decreased the levels of TNFα and particularly those containing ketorolac, supporting its potent anti-inflammatory properties. These findings support the use of ketorolac in combination with a hydrogel formulation as a potential effective therapy for retinal diseases. As future recommendation, in vivo studies in animal models of retinal degeneration are crucial to confirm the security and efficacy of this novel therapy.

## 5. Conclusions

Ultimately, our work introduces six different fully developed biodegradable thermo-responsive injectable hydrogel formulations, with preliminary evaluation of their in vitro tolerance and efficacy (anti-inflammatory and antioxidant activity) in retinal pigmented epithelial cells. These novel platforms for neuroprotective combined therapy were shown to be well tolerated in retinal cells and also showed robust, long-lasting and controlled in vitro release profiles overtime. These formulations could entail a new generation of sustained neuroprotective treatments that could improve the efficacy of current treatments and the quality of life of patients.

## Figures and Tables

**Figure 1 pharmaceutics-13-00234-f001:**
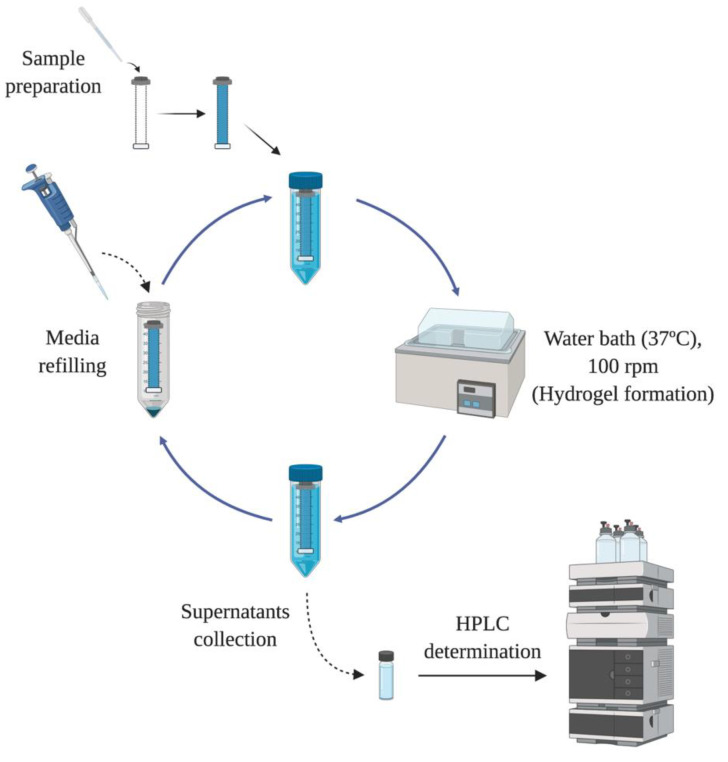
Design of the in vitro release experiments for dexamethasone and ketorolac.

**Figure 2 pharmaceutics-13-00234-f002:**
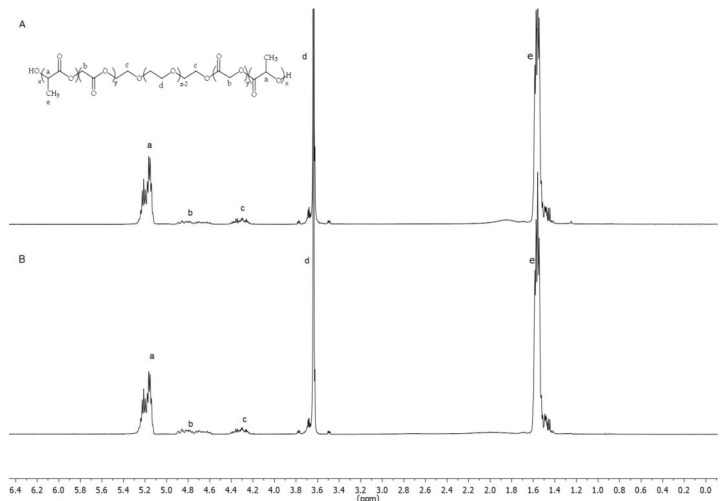
Nuclear magnetic resonance (NMR) spectrum of copolymer 1 and 2 (**A**,**B**) showing their chemical structure.

**Figure 3 pharmaceutics-13-00234-f003:**
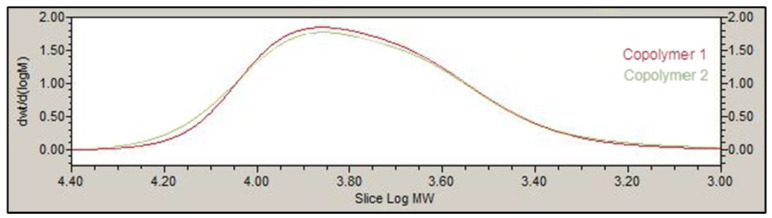
Gel permeation chromatography (GPC) dwt/d(logM) vs. log MW showing the molecular weight distribution for copolymers 1 and 2.

**Figure 4 pharmaceutics-13-00234-f004:**
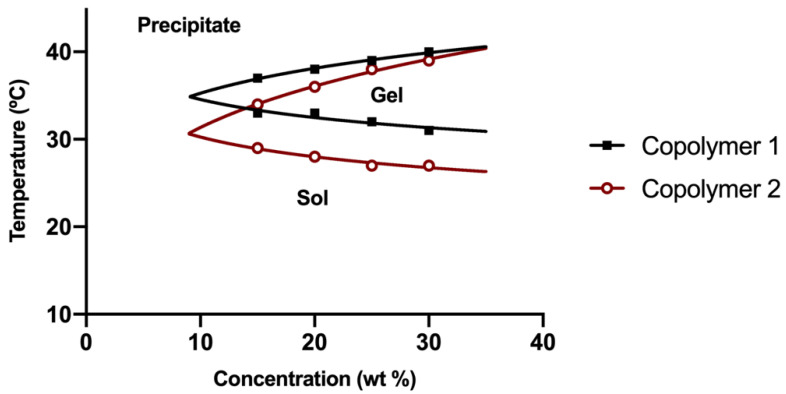
Phase diagram of copolymer 1 (HyG-1) and 2 (HyG-2) in the final bicarbonate buffer.

**Figure 5 pharmaceutics-13-00234-f005:**
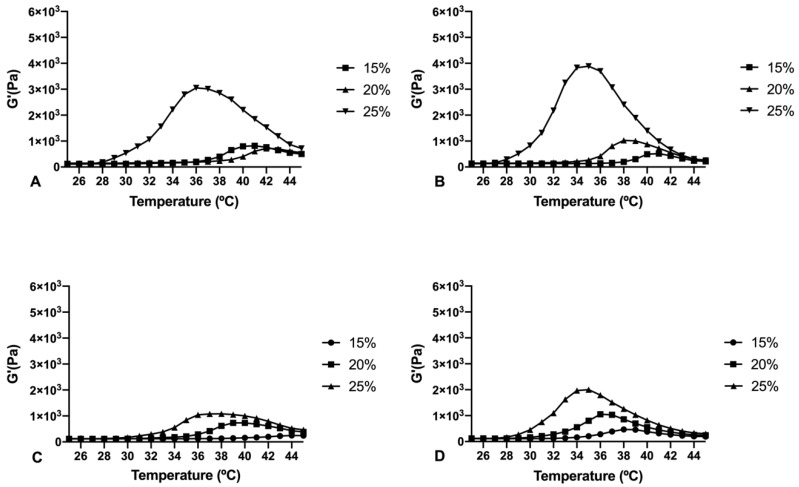
Rheological behaviour values (G’ modulus) of PLGA-PEG-PLGA HyG-1 at different concentrations dispersed in water (**A**) and the bicarbonate buffer (**B**) and HyG-2 hydrogels also dispersed in water (**C**) and the bicarbonate buffer (**D**).

**Figure 6 pharmaceutics-13-00234-f006:**
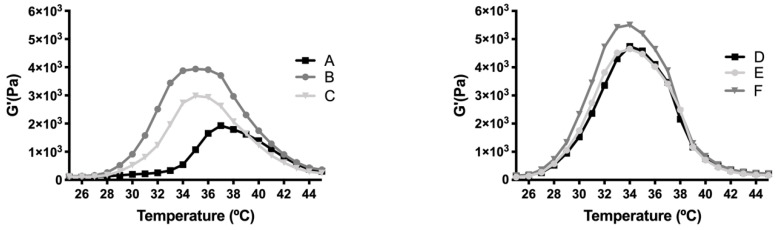
Rheology of Dexamethasone (**left**) and Ketorolac (**right**) formulations with selected HyG-1 in bicarbonate buffer at 25% of PLGA-PEG-PLGA. DX 0.2% (A), DX 0.2% + Idebenone 1 µM (B), DX 0.2% + TPGS 0.02% (C), KT 0.5% (D), KT 0.5% + Idebenone 1 µM (E) and KT 0.5% + TPGS 0.02% (F).

**Figure 7 pharmaceutics-13-00234-f007:**
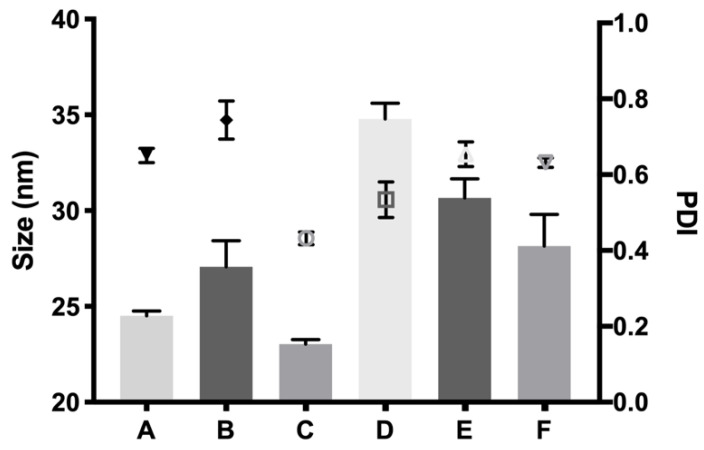
Size of the micelles in nanometers in comparison with the single icons indicating the polydispersity index of sizes (PDI) of DX 0.2% (A), DX 0.2% + Idebenone 1 µM (B), DX 0.2% + TPGS 0.02% (C), KT 0.5% (D), KT 0.5% + Idebenone 1 µM (E) and KT 0.5% + TPGS 0.02% (F), respectively.

**Figure 8 pharmaceutics-13-00234-f008:**
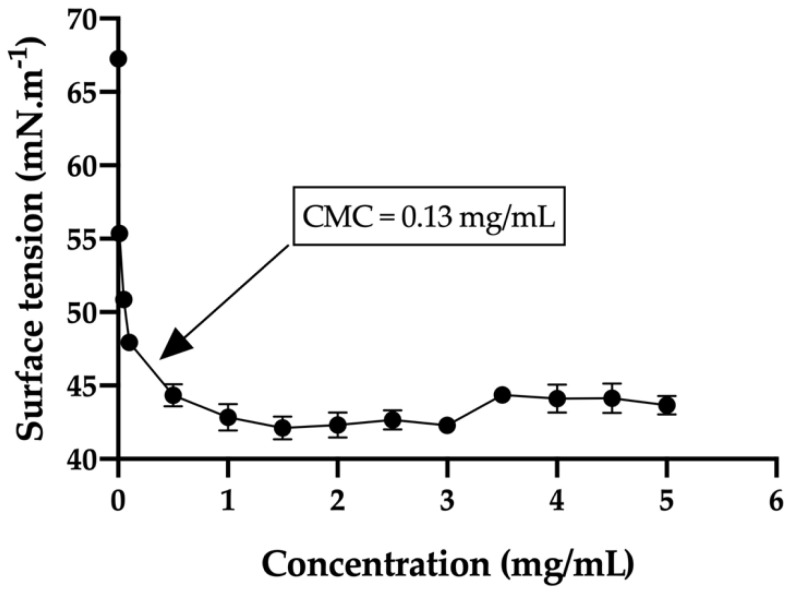
Surface tension diagram of the selected copolymer 1 showing the critical micelle concentration (CMC).

**Figure 9 pharmaceutics-13-00234-f009:**
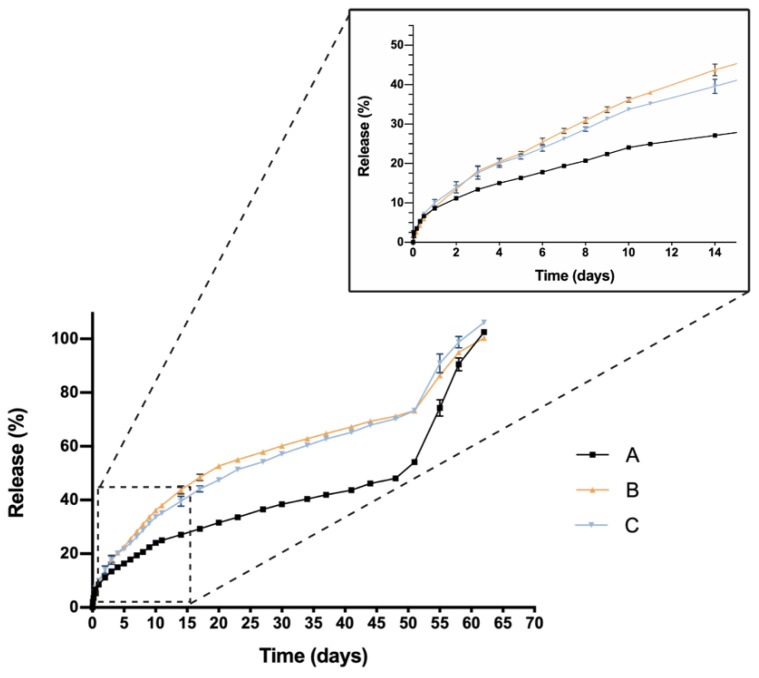
In vitro drug release profile of dexamethasone-based formulations with TPGS or Idebenone respectively. DX 0.2% (A), DX 0.2% + Idebenone 1 µM (B), DX 0.2% + TPGS 0.02% (C). Data are represented as mean ± standard deviation (SD) from 3 different batches (*n* = 3).

**Figure 10 pharmaceutics-13-00234-f010:**
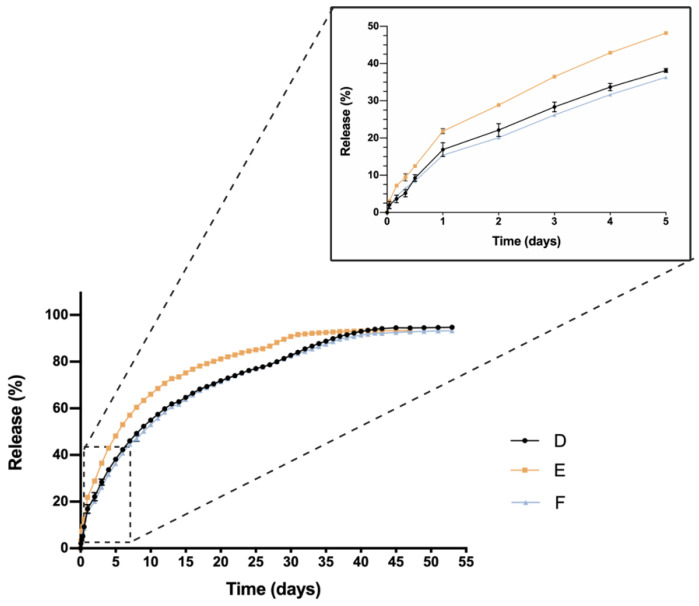
In vitro drug release profile of ketorolac-based formulations with TPGS or Idebenone respectively; KT 0.5% (D), KT 0.5% + Idebenone 1 µM (E) and KT 0.5% + TPGS 0.02% (F). Data are represented as mean ± standard deviation (SD) from three different batches (*n* = 3).

**Figure 11 pharmaceutics-13-00234-f011:**
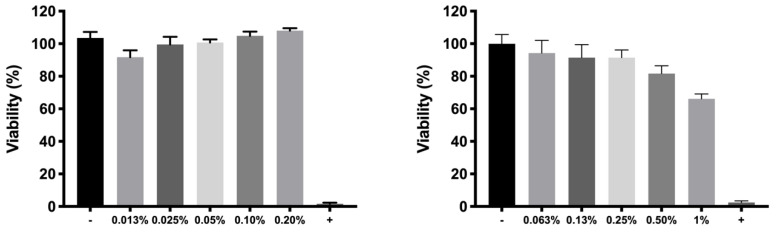
Cell viability of dexamethasone phosphate (**left**) and ketorolac tris salt (**right**) at different concentrations in RPE-1 cells. Data are expressed as mean ± standard deviations. Cell viability (%) is calculated in relation to the negative control (100% of viability).

**Figure 12 pharmaceutics-13-00234-f012:**
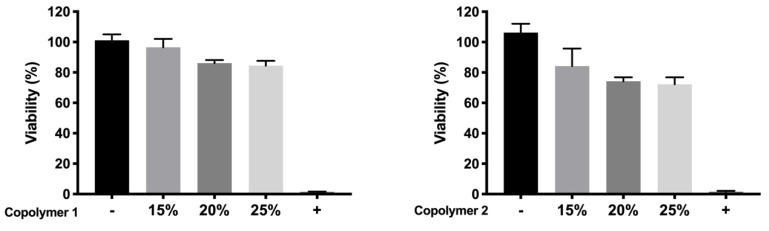
Viability of copolymers 1 and 2 respectively in bicarbonate buffer without drugs.

**Figure 13 pharmaceutics-13-00234-f013:**
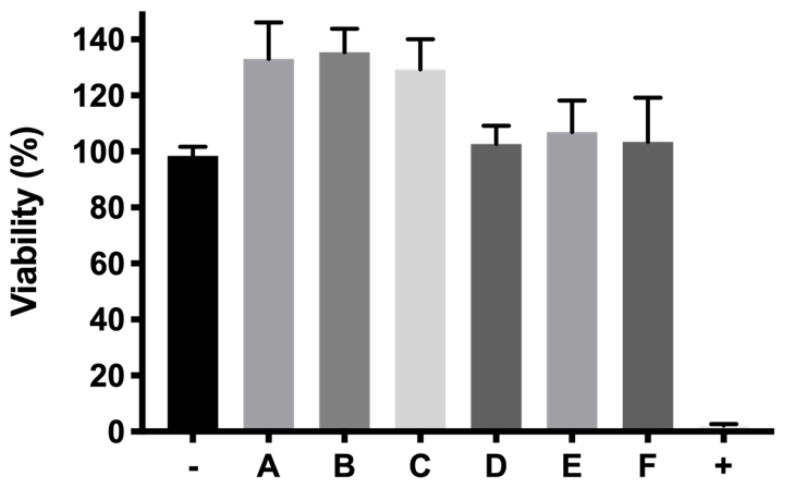
Viability of final formulations (%) prepared with copolymer 1. DX 0.2% (A), DX 0.2% + Idebenone 1 µM (B), DX 0.2% + TPGS 0.02% (C), KT 0.5% (D), KT 0.5% + Idebenone 1 µM (E) and KT 0.5% + TPGS 0.02% (F), respectively.

**Figure 14 pharmaceutics-13-00234-f014:**
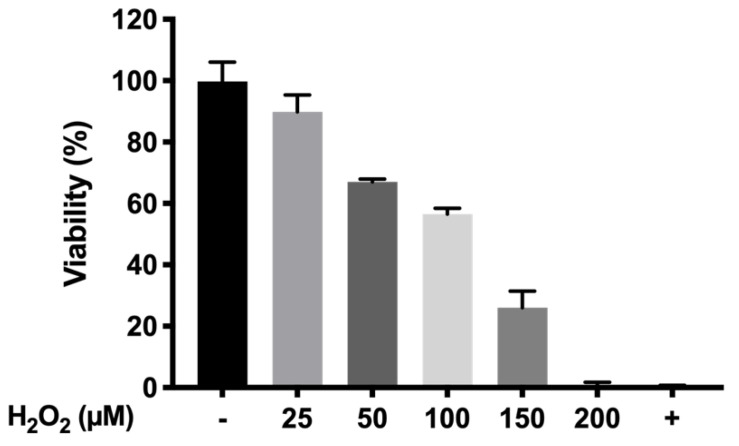
Optimization of cell viability in the oxidative stress model at different H_2_O_2_ concentrations.

**Figure 15 pharmaceutics-13-00234-f015:**
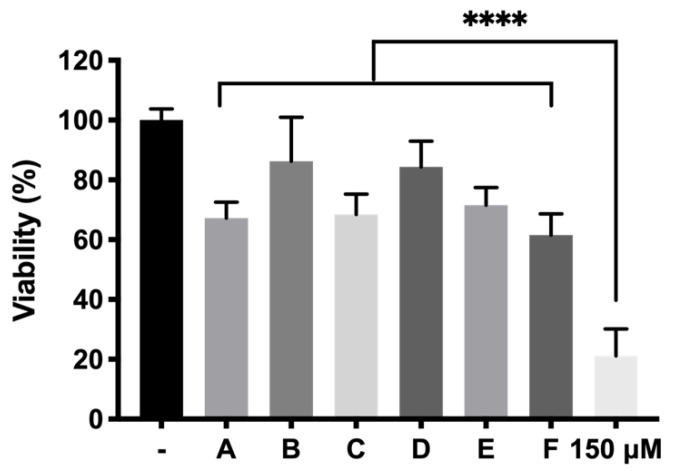
In vitro protection of the final formulations DX 0.2% (A), DX 0.2% + Idebenone 1 µM (B), DX 0.2% + TPGS 0.02% (C), KT 0.5% (D), KT 0.5% + Idebenone 1 µM (E) and KT 0.5% + TPGS 0.02% (F), respectively, in response to 150 µM of (positive control, +) oxidative stress in RPE-1 cells. High statistically significant values (*p* ≤ 0.0001; ****).

**Figure 16 pharmaceutics-13-00234-f016:**
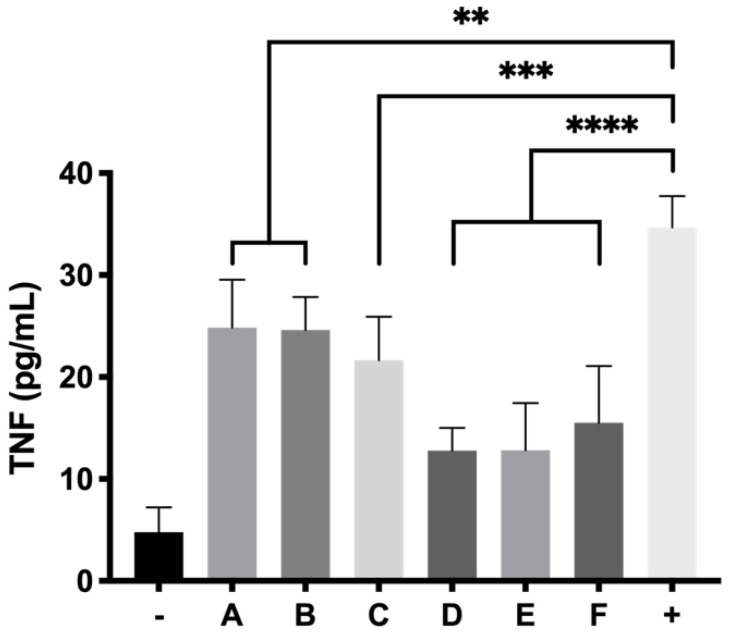
TNFα activity of RPE-1 cells exposed to final formulations DX 0.2% (A), DX 0.2% + Idebenone 1 µM (B), DX 0.2% + TPGS 0.02% (C), KT 0.5% (D), KT 0.5% + Idebenone 1 µM (E) and KT 0.5% + TPGS 0.02% (F), respectively, in response to LPS induced inflammation. *p* ≤ 0.01 (**), *p* ≤ 0.001 (***) and *p* ≤ 0.0001 (****).

**Table 1 pharmaceutics-13-00234-t001:** Composition of formulations containing the combined therapy.

Formulation	Polymer Concentration	GA:LA Ratio	Dexamethasone	Ketorolac	Idebenone	Tocopherol Polyethylene Glycol 1000 Succinate (TPGS)
A	25%	1:15	0.2%	-	-	-
B			0.2%	-	1 µM	-
C			0.2%	-	-	0.002%
D			-	0.5%	-	-
E			-	0.5%	1 µM	-
F			-	0.5%	-	0.002%

**Table 2 pharmaceutics-13-00234-t002:** GPC and NMR copolymers’ characterization.

Polymers Synthesized	Theoretical Ratio (PEG:GA:LA)	Theoretical Ratio (GA:LA)	NMR Experimental Ratio (PEG:GA:LA)	NMR Experimental Ratio (GA:LA)	Mn ^1^	Mn ^2^	Mw ^2^	PDI ^3^
Copolymer 1	1:1.54:23.10	1:15	1:1.51:20.19	1:14	4585.25	5377.70	6581.35	1.22
Copolymer 2	1:2.25:22.50	1:10	1:1.83:17.17	1:9.80	4235.87	4994.73	6792.03	1.36

^1^ Molecular weight (Mw) calculated by ^1^H-NMR. ^2^ Number average Mw and weight average Mw obtained by GPC. ^3^ Polydispersity (Mw/Mn) calculated by GPC.

**Table 3 pharmaceutics-13-00234-t003:** Kinetic parameter and release mechanism proposed.

Hydrogels	R^2^	*n* Value	Mechanism
A	0.9981	0.45	Fickian diffusion
B	0.9969	0.62	Anomalous transport
C	0.9988	0.52	Anomalous transport
D	0.9891	0.63	Anomalous transport
E	0.9970	0.63	Anomalous transport
F	0.9966	0.57	Anomalous transport

## Data Availability

The data presented in this study are available on request from the corresponding author.

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
