# Peer review of "Thermo-Responsive PLGA-PEG-PLGA Hydrogels as Novel Injectable Platforms for Neuroprotective Combined Therapies in the Treatment of Retinal Degenerative Diseases"

_pharmaceutics, 2021, doi:10.3390/pharmaceutics13020234_

Round 1

Reviewer 1 Report

  1. What is the potential in vivo toxicity of the thermo-responsive PLGA-PEG-PLGA hydrogels?
  2. What is the potential in vivo immunologic response of these hydrogels? In particular, do the investigators anticipate an allergic response in human subjects with a PEG allergy?
  3. In the final sentence, the authors state, "As future recommendations, in vivo studies in animal models of retinal degeneration are crucial to confirm the security and efficacy of this novel therapy." Which specific animals do they anticipate utilizing for their first in vivo studies?

Author Response

Reviewer 1

We would like to thank Reviewer 1 for the time and effort he/she has dedicated to our manuscript. We would like to answer him/her question by question

1. What is the potential in vivo toxicity of the thermo-responsive PLGA-PEG-PLGA hydrogels?

We estimate that the PLGA-PEG-PLGA will have a suitable in vivo tolerance since a study from Gao. Y and co-workers [1] supports the use of PLGA-PEG-PLGA triblock in vivo. They studied the blood compatibility of the hydrogels by hemolysis test and was fully compatible. They also did viability studies in chicken embryonic fibroblast and chicken macrophage-like cells. Besides, they extracted tissue samples from the site of injection (muscle and skin) and did not observe any inflammatory response nor cell infiltration through hematoxylin-eosin (HE) staining assay. Although we are not aware of in vivo studies of intravitreal injections, this could be a good starting point to foresee future results in vivo, considering also that it is well-known that both PLGA and PEG separately are well tolerated after intravitreal administration. We would like to thank the reviewer for this comment, and we will take it into consideration for future in vivo studies

2. What is the potential in vivo immunologic response of these hydrogels? In particular, do the investigators anticipate an allergic response in human subjects with a PEG allergy?

Immunogenicity of PLGA-PEG-PLGA hydrogels is a very interesting and important concern. We would like to thank this reviewer for underlaying it. As reported in question 1, previous studies suggest that PLGA-PEG-PLGA based hydrogels could be well tolerated without showing dangerous immunologic responses. Regarding subjects presenting PEG allergy, we speculate that some level of allergic reaction could be given, although we foresee that hydrogels would not pass to systemic circulation and risks of reaction could be limited. However, it is important to remark that this particular question should be assed and taken into consideration for further study.

3. In the final sentence, the authors state, "As future recommendations, in vivo studies in animal models of retinal degeneration are crucial to confirm the security and efficacy of this novel therapy." Which specific animals do they anticipate utilizing for their first in vivo studies?

The selection of an animal model for studying a disease is a decision that must be very carefully chosen, since it can provide researchers with useful and interesting data in the way of translating a therapy to humans. The P23H model with rats or knock-out mice would be a good starting point. It has been observed that these animals possess retinas that have many similarities compared to humans. Besides, the accelerated lifespan of mice (30 days) allows researchers to study the natural progression of certain diseases and their treatments [2]. Another study shows an interesting model induced with sodium iodate injection. It produces damage, apoptosis of photoreceptors and an extensive DNA damage in the retina [3].

References

  1. Gao, Y.; Ji, H.; Peng, L.; Gao, X.; Jiang, S. Development of PLGA-PEG-PLGA Hydrogel Delivery System for Enhanced Immunoreaction and Efficacy of Newcastle Disease Virus DNA Vaccine. Molecules 2020, 25, 1–15.
  2. Chang, B. Mouse models for studies of retinal degeneration and diseases. Methods Mol. Biol. 2013, 935, 27–39.
  3. Koh, A.E.H.; Alsaeedi, H.A.; Rashid, M. binti A.; Lam, C.; Harun, M.H.N.; Saleh, M.F. bin M.; Luu, C.D.; Kumar, S.S.; Ng, M.H.; Isa, H.M.; et al. Retinal degeneration rat model: A study on the structural and functional changes in the retina following injection of sodium iodate. J. Photochem. Photobiol. B Biol. 2019, 196, 111514.

Reviewer 2 Report

The manuscript by López-Cano et al. describes development, drug loading, characerization and in vitro test of new hydrogels for treatment of retinal neurodegenerative diseases. The approach is rigorous, the methods well described, and my overall evaluation of the manuscript is highly positive. I have only a small number of minor concerns that should be addressed before the manuscript is publishable.

1) introduction is reasonably focused on retinal pathologies. However, in my opinion is far too long. It resembles more an extensive review and should be summarized for clarity and ease of reading.

2) figure 1 is not necessary.

3) please merge the two plots in figure 3, and significantly enlarge characters. As presented, the figure is nearly unreadable.

4) please merge figures 5-6 into a single one

5) I have some doubts regarding micelles PDI. 0.4-0.6 is quite a large value in nanostructures. How these values compare with other reported similar structures? Could it be related to the presence of a small fraction of aggregates with significantly larger size? please comment.

6) Release experiments: error bars are missing in most points. Please add them. It is not clear to me how the different release profile at day 51 is interpreted. Could the authors better clarify this aspect?

Author Response

Reviewer 2

The manuscript by López-Cano et al. describes development, drug loading, characterization and in vitro test of new hydrogels for treatment of retinal neurodegenerative diseases. The approach is rigorous, the methods well described, and my overall evaluation of the manuscript is highly positive. I have only a small number of minor concerns that should be addressed before the manuscript is publishable.

We would like to thank Reviewer 2 for the time and effort he/she has dedicated to our manuscript. We would like to answer him/her question by question

1) introduction is reasonably focused on retinal pathologies. However, in my opinion is far too long. It resembles more an extensive review and should be summarized for clarity and ease of reading.

According to the reviewer’s comment we have reduced the extension of the introduction. We would like to thank this helpful recommendation

2) figure 1 is not necessary.

We fully appreciate your suggestion; however, we reckon that the figure could help the reader to better understand the experimental view of the in vitro release experiment and therefore increase its reproducibility.

3) please merge the two plots in figure 3, and significantly enlarge characters. As presented, the figure is nearly unreadable.

We would like to thank reviewer 2 for his/her suggestion. Accordingly, Figure 3 has been changed so it can be better read with enlarged and more clear letters, and the plots have been overlaid. 

4) please merge figures 5-6 into a single one

As kindly suggested, the figures 5 and 6 have been merged into a single one (now figure 5).

5) I have some doubts regarding micelles PDI. 0.4-0.6 is quite a large value in nanostructures. How these values compare with other reported similar structures? Could it be related to the presence of a small fraction of aggregates with significantly larger size? please comment.

We would like to thank the reviewer for this interesting remark. According to some researched bibliography, we can say that polymers, and especially triblock polymers with high molecular weights and some level of branching, could derive in micelles with some higher PDIs, but still stable in aqueous solutions. For example, Lee K et al. (reference 44 in the text) obtains polymer micelles with PDI between 0.026 and 0.777 from mixing by esterification α-Tocopheryl succinate (α-TOS) molecules and a mPEG-b-PHEMA hydrophilic diblock copolymer. In another recent publication Gao Y and co-workers [1] developed a PLGA-PEG triblock as a novel delivery system for a DNA vaccine and obtain a micelles PDI of 0.249. However, it is worth mentioning that they used lower concentrations of the copolymer and that the GA/LA ratios differed very much from ours, being their LA ratio 5-folds smaller than our LA ratio. That could derive in lower molecular weight copolymers and micelle size could change. However, in the text the phrase low PDIs has been changed, stating “PDI are shown” instead of “low PDIs”.

As the reviewer very interestingly suggests, some type of aggregation could be given due to the relatively high viscosity of the sample. This idea has been also included in the text

6) Release experiments: error bars are missing in most points. Please add them. It is not clear to me how the different release profile at day 51 is interpreted. Could the authors better clarify this aspect?

We would like to thank this reviewer’s comment because it gives us the opportunity to clarify that the error bars have not been removed from the in vitro release curves. It is worth mentioning here that all the results are expressed as x̄ ± sd (n= 3). We have also included it in the description below in figures 9 and 10 to help reader’s comprehension. However, the deviations were so small in some points that cannot be appreciated, although they can be in other regions of the release profile. To enhance the reader comprehension, we have smoothed the bars and icons of the graphs so they can be better appreciated but are still the same.

Regarding the release profile at day 51, we state that a first phase of sustained release sustained release was observed until that day since in the three cases an increment in slope was observed. As shown below, A goes from 54% to 74% and B and C from 73% to around 86 and 90% respectively.

References

  1. Gao, Y.; Ji, H.; Peng, L.; Gao, X.; Jiang, S. Development of PLGA-PEG-PLGA Hydrogel Delivery System for Enhanced Immunoreaction and Efficacy of Newcastle Disease Virus DNA Vaccine. Molecules 2020, 25, 1–15.

Reviewer 3 Report

The authors investigated the properties of thermoresponsive injectable hydrogel that should be capable of delivering drugs to the retina. The authors performed a great deal of work. There are some comments to be made.

There are several minor inaccuracies in the Introduction. Course of age-related macular degeneration (AMD) is not described correctly, and wording of “dry” and “wet” AMD is usually avoided nowadays. The statement “pathophysiology of RP is still unknown” is surprising, as there has been done a lot of research on this field. Most of RP forms are monogenetic, by the way. Please explain the sentence in lines 89/90 “As some of the diseases mentioned above, RP experiments RPE degeneration and cell death.” In general, the reviewer does not see why in the manuscript a whole page is used to describe different ocular diseases. Maybe it is enough to state that several ocular diseases are connected with oxidative stress and inflammation, and immunosuppressive and antioxidative drugs are recommended.

Inscription of most diagram axes is rather small, and sometimes the lines are really thin (e.g., in Figure 3). Do the authors see the opportunity to use larger letters and numbers, and to use thicker lines?

In Figure 8, the authors should not use a line to connect dots showing the polydispersity index, as there is no direct connection between the single groups.

In the lines 427/428, the authors mention possible erosion of the hydrogel. This leads to the question of long-term stability of the hydrogels. Do the authors any information about it?

In Figs. 12A and 12B, is there a rationale to display the results from the highest used concentrations to the lowest? The reviewer finds it somehow confusing. Moreover, it is not the logical order assuming that the concentration of the control on the left-hand side is zero. The same question arises with Figure 15.

Please check in line 697 what you do mean by “Authors, also discovered…”.

Finally, some language polishing is highly recommended. For instance, use of commas is not correct sometimes, and the word “respectively” should be used with care.

Author Response

Reviewer 3

The authors investigated the properties of thermoresponsive injectable hydrogel that should be capable of delivering drugs to the retina. The authors performed a great deal of work.

We would like to thank Reviewer 3 for the time and effort he/she has dedicated to our manuscript. We would like to answer him/her question by question

There are some comments to be made. There are several minor inaccuracies in the Introduction. Course of age-related macular degeneration (AMD) is not described correctly, and wording of “dry” and “wet” AMD is usually avoided nowadays. The statement “pathophysiology of RP is still unknown” is surprising, as there has been done a lot of research on this field. Most of RP forms are monogenetic, by the way. Please explain the sentence in lines 89/90 “As some of the diseases mentioned above, RP experiments RPE degeneration and cell death.”

We would like to thank reviewer 3 for their interesting and helpful comments.

The AMD description has been shortened according to the introduction reduction and updated according to the reviewer suggestion. Besides, changes in fundus structures, neovascularization, exudative and atrophy have been included.

When we claimed that “pathophysiology of RP is still unknown” we meant that although a lot of research has been conducted there is not a single mechanisms that could explain it as stated by some authors [1]. This sentence has been changed by “RP pathophysiology has been widely studied”.

The sentence “As some of the diseases mentioned above, RP experiments RPE degeneration and cell death.” has been corrected to “As some of the diseases mentioned above, in RP, patients experiment RPE degeneration progressing to cell death”

In general, the reviewer does not see why in the manuscript a whole page is used to describe different ocular diseases. Maybe it is enough to state that several ocular diseases are connected with oxidative stress and inflammation, and immunosuppressive and antioxidative drugs are recommended.

We would like to thank review 3 for his/her kind suggestion. The first part of the introduction has been reduced accordingly

Inscription of most diagram axes is rather small, and sometimes the lines are really thin (e.g., in Figure 3). Do the authors see the opportunity to use larger letters and numbers, and to use thicker lines?

We thank the reviewer for the consideration since it will enhance the reader experience. We have used larger letter for almost all the graphs included the figure 3, that has been modified and overlaid (as requested by one of the reviewers).

In Figure 8, the authors should not use a line to connect dots showing the polydispersity index, as there is no direct connection between the single groups.

Figure 8 (now figure 7 due to other figures corrections) has been modified and polydispersity points are not connected anymore.

In the lines 427/428, the authors mention possible erosion of the hydrogel. This leads to the question of long-term stability of the hydrogels. Do the authors any information about it?

The importance of long-term stability is a very interesting and an important point to bear in mind. These polymers have been specifically designed to obtain biodegradable hydrogels that disappear in an aqueous solution overtime. The authors comment the different mechanisms by which the hydrogels are able to release the active compounds. Among them, erosion of the hydrogel is one of the proposed mechanisms when the hydrogel is created at physiological temperature, since PLGA-PEG-PLGA micelles will be degrading and aqueous dispersant will be eroding the gel between others.

The freeze-dried polymer can be storage at -30ºC for very long term protected from moisture. Besides in aqueous solution it can be storage at 2-8ºC up to one month (refrigerated in sol state) and can still form hydrogels at physiological temperatures. Stability in aqueous solution at room temperature is an important feature that will be developed as a future prospect.

In Figs. 12A and 12B, is there a rationale to display the results from the highest used concentrations to the lowest? The reviewer finds it somehow confusing. Moreover, it is not the logical order assuming that the concentration of the control on the left-hand side is zero. The same question arises with Figure 15.

We would like to thank the reviewer’s suggestion. In fact, the way the reviewer has exposed the display of the results in these figures makes much more sense so the concentrations can be studied from 0 (nontoxic) to highly toxic ones. The figure has been properly changed.

Please check in line 697 what you do mean by “Authors, also discovered…”.

The sentence meant to explain that these authors also reported a protective roll of dexamethasone against ROS species. The sentence has been corrected by “Besides, they also reported that dexamethasone protected against ROS and increased survivability of RPE cells and photoreceptors”.

Finally, some language polishing is highly recommended. For instance, use of commas is not correct sometimes, and the word “respectively” should be used with care.

We would like to thank reviewer 3 for his/her help, we have carefully revised grammar all around the text.

References

  1. Park, U.C.; Park, J.H.; Ma, D.J.; Cho, I.H.; Oh, B.L.; Yu, H.G. A randomized paired-eye trial of intravitreal dexamethasone implant for cystoid macular edema in retinitis pigmentosa. Retina 2020, 40, 1359–1366.

Reviewer 4 Report

This is a good manuscript that is clearly written an designed.

I recommend its publication after addressing the following points:

1- Did the authors made any compatibility studies between TPGS and Idebenone with the investigated drugs? 

2- How do the authors explain the high cytotoxicity effect obtained from the 1% ketorolac tris salt reaching a viability of almost 65% only.

3- Please specify the nature of the error bars of all of the demonstrated figures.

Author Response

Reviewer 4

This is a good manuscript that is clearly written and designed.

I recommend its publication after addressing the following points:

We would like to thank Reviewer 4 for the time and effort he/she has dedicated to our manuscript. We would like to answer him/her question by question

1- Did the authors made any compatibility studies between TPGS and Idebenone with the investigated drugs? 

For this specific work compatibility studies were not performed, apart from demonstrating that the combination of TPGS or Idebenone with the active ingredients did not affect the cells viability and that they are effective. However, this is a very interesting suggestion for future studies

2- How do the authors explain the high cytotoxicity effect obtained from the 1% ketorolac tris salt reaching a viability of almost 65% only.

We speculate that some level of concentration between 1% and 0.5% can be a threshold that drastically decrease cel viability. Another reason would be the acidity in solution, since ketorolac presents a pKa value of 3.84 and concentration could increase the acidity making the cells more susceptible to die. Nevertheless, we confirm that the concentration used for the article (0.5%) is established as safe and that the formulations not only are not toxic but protective.

3- Please specify the nature of the error bars of all of the demonstrated figures.

We would like to thank the reviewer’s comment. It is worth mentioning here that all the results are expressed as x̄ ± sd (n= 3). We have also included it in the description below in figures 9 and 10 to help reader’s comprehension. We also would like to clarify that the error bars have not been removed from the in vitro release curves. However, the deviations were so small in some points that cannot be appreciated, although they can be in other regions of the release profile. To enhance the reader comprehension, we have smoothed the bars and icons of the graphs so they can be better appreciated but are still the same.
